# Early Detection of Therapeutic Benefit from PD-1/PD-L1 Blockade in Advanced Lung Cancer by Monitoring Cachexia-Related Circulating Cytokines

**DOI:** 10.3390/cancers15041170

**Published:** 2023-02-11

**Authors:** Shiting Xu, Keita Miura, Takehito Shukuya, Sonoko Harada, Masahiro Fujioka, Wira Winardi, Shoko Shimamura, Kana Kurokawa, Issei Sumiyoshi, Taichi Miyawaki, Tetsuhiko Asao, Yoichiro Mitsuishi, Ken Tajima, Fumiyuki Takahashi, Takuo Hayashi, Norihiro Harada, Kazuhisa Takahashi

**Affiliations:** 1Department of Respiratory Medicine, Graduate School of Medicine, Juntendo University, 3-1-3 Hongo, Bunkyo-ku, Tokyo 113-8421, Japan; 2Atopy (Allergy) Research Center, Graduate School of Medicine, Juntendo University, Bunkyo-ku, Tokyo 113-8421, Japan; 3Department of Diagnostic Pathology, Graduate School of Medicine, Juntendo University, Bunkyo-ku, Tokyo 113-8421, Japan

**Keywords:** cancer cachexia, circulating cytokines, lung cancer, immune checkpoint therapy, prognosis

## Abstract

**Simple Summary:**

Cancer cachexia is a metabolic disorder that is associated with poor immunotherapeutic outcomes. However, the circulating cachexia-related cytokines have yet to be longitudinally examined to assess their role in predicting the therapeutic outcomes of PD-1/PD-L1 blockade in advanced lung cancer. This prospective study identified cachexia-related cytokines from a panel of 41 circulating cytokines, which were examined at baseline and during treatment. Our study showed that high IL-6 was associated with a higher risk of immune-related adverse events, while high IL-10 was associated with poor overall survival. More importantly, our study revealed for the first time that an early increase in eotaxin-1 after immunotherapy is a favorable factor related to the therapeutic response to PD-1/PD-L1 blockade and overall survival. These results suggested that the blood-based evaluation of a cachexia-related cytokine network may provide early clues for the immunotherapeutic outcomes.

**Abstract:**

Cancer cachexia is associated with poor immunotherapeutic outcomes. This prospective observational study longitudinally evaluated the role of cachexia-related circulating cytokines in predicting the risk and benefit of PD-1/PD-L1 blockade in advanced lung cancer. Forty-one circulating cytokines at baseline and after one cycle of PD-1/PD-L1 blockade treatment were measured in patients with advanced lung cancer between 2019 and 2020. The cachexia-related cytokines were identified by comparing the levels of circulating cytokines between cachectic and non-cachectic patients. Among 55 patients, 49.1% were diagnosed with cachexia at the beginning of PD-1/PD-L1 blockade therapy. Baseline levels of the circulating cytokines IL-6, IL-8, IL-10, IL-15, and IP-10 were significantly higher in cachectic patients. In contrast, the level of eotaxin-1 was lower in cachectic patients than in those without cachexia. Higher IL-6 at baseline and during treatment was associated with a greater risk of immune-related adverse events, while higher IL-10 at baseline was linked to worse overall survival. More importantly, increased eotaxin-1 after one cycle of PD-1/PD-L1 blockade treatment was associated with higher objective response and better overall survival. A blood-based, cachexia-related cytokine assay may yield potential biomarkers for the early prediction of clinical response to PD-1/PD-L1 blockade and provide clues for improving the outcomes of cachectic patients.

## 1. Introduction

Lung cancer treatment has been altered radically by the advent of immune checkpoint therapy over the last few years [1]. Despite programmed cell death protein 1/ligand 1 (PD-1/PD-L1) blockade significantly improving the survival of patients with advanced lung cancer, a certain number of patients fail to respond to PD-1/PD-L1 blockade therapy, or eventually progress after an initial response [2,3]. In order to select suitable patients, previous studies mainly focused on evaluating tumor-derived factors such as PD-L1 expression on tumor cells, tumor mutation burden, and the tumor microenvironment [4]. However, PD-1/PD-L1 blockade enhances the killing of tumor cells by stimulating the host anti-tumor immune system [5]. Therefore, host immunity and nutrition play important roles in PD-1/PD-L1 blockade therapy.

Cachexia, characterized by weight loss and disordered immunonutrition, accounts for a larger proportion of patients with advanced lung cancer [6]. Cancer cachexia involves the diverse crosstalk of mediators arising from the cancer cells and cells within the tumor microenvironment, including inflammatory and immune cells [7,8]. Emerging evidence showed that immunometabolic alterations in the context of cancer cachexia are closely associated with poor immunotherapeutic outcomes in cancers [9,10]. Cachexia-related inflammatory cytokines are reported to include IL-1, IL-6, TNF-α, and IFN-γ, which adversely influence systemic disorders such as metabolic abnormalities, skeletal muscle loss, and fat breakdown [11]. In advanced cancer, the numerous cytokines act as immunological signaling proteins primarily at a local cellular level, but are also released into the circulation. As elements of a complex immune and metabolic response leading to cachexia, there is an interplay between inflammatory cytokines and the antitumor activity of host immune cells [12,13]. However, many of the cytokines that are implicated in cachexia have not been systematically characterized.

Currently, the underlying relationship between circulating cachexia-related cytokines and therapeutic outcomes remains unclear, including the risk and benefit of PD-1/PD-L1 blockade. This study was designed to examine the circulating inflammatory cytokines before PD-1/PD-L1 blockade therapy and after one cycle of PD-1/PD-L1 blockade therapy, in order to identify cachexia-related cytokines and evaluate their clinical relevance after treatment with PD-1/PD-L1 blockade.

## 2. Materials and Methods

### 2.1. Patient Characteristics

We prospectively enrolled all consecutive patients with advanced lung cancer who underwent PD-1/PD-L1 blockade, either as monotherapy or combined with platinum-based chemotherapy, from July 2019 to December 2020.

The study recorded data for pre-treatment patient demographics (sex, age, smoking history, body weight, and body weight change during the previous six months), Eastern Cooperative Oncology Group (ECOG) performance status, and peripheral blood data before PD-1/PD-L1 blockade therapy including albumin (Alb), hemoglobin (Hb), C-reactive protein (CRP), and neutrophil-to-lymphocyte ratio (NLR). Tumor characteristics including histology, tumor molecular profiling for *EGFR* mutation, *ALK* fusion, *BRAF* mutation and *ROS1* fusion, PD-L1 status, tumor–node–metastasis (TNM) classification as proposed by the 8th edition of Union for International Cancer Control (UICC), and the number of metastatic organs. Treatment data including the ordinal line of treatment, type of immunotherapy, immune-related adverse events, therapeutic response, and patient survival were also collected.

### 2.2. Definition of Cancer Cachexia

Cachexia was defined as weight loss > 5% within the six months prior to initial PD-1/PD-L1 blockade [9,14]. Consequently, patients were classified as either ‘cachexia’ or ‘non-cachexia’ to allow the identification of cachexia-related inflammatory cytokines.

### 2.3. Blood Collection

Peripheral blood was collected from consenting patients before (baseline) and after the first cycle, that is, after three weeks of PD-1/PD-L1 blockade therapy (during treatment). The final concentration of heparin was approximately 15 USP per mL of blood. Plasma was produced by centrifugation at 500× *g* for 10 min, and then aliquoted and stored at −80 °C until batch analysis.

### 2.4. Multiplex Analysis

The plasma levels of 41 inflammatory cytokines/chemokines (interleukin (IL)-1 alpha, IL-1 beta, IL-1 R alpha, IL-2, IL-3, IL-4, IL-5, IL-6, IL-8, IL-9, IL-10, IL-12(p40), IL-12(p70), IL-13, IL-15, IL-17A/CTLA8, sCD40L, EGF, eotaxin-1/CCL11, FGF-1/FGF-basic, Flt3-ligand, Fractalkine/CX3CL1, G-CSF, GM-CSF, GRO, IFN alpha2, IFN- gamma, IP-10/CXCL10, MCP-1/CCL2, MCP-3/CCL7, MDC/CCL22, MIP-1 alpha/CCL3, MIP-1 beta/CCL4, PDGF-AA, PDGF-AB/BB, RANTES/CCL5, TGF alpha, tumor necrosis factor [TNF]-alpha, TNF-beta, VEGF-A) were conducted using the MILLIPLEX^®^ MAP Human Cytokine/Chemokine Magnetic Bead Panel Kit (Merck Millipore, Burlington, MA, USA). The multiplex assay was conducted using the Luminex^®^ 200TM System (Luminex Corporation, Austin, TX, USA) according to the manufacturer’s instructions. Standard curves were generated using the specific standards supplied by the manufacturer.

### 2.5. Therapeutic Efficacy and Survival Analysis

Immune-related adverse events (irAEs) were graded according to the Common Terminology Criteria for Adverse Events version 5.0 [15]. The tumor responses were classified as complete response (CR), partial response (PR), stable disease (SD), and progressive disease (PD) according to the response evaluation criteria in solid tumors (RECIST), version 1.1 [16]. Follow-up evaluation was initiated on the date of PD-1/PD-L1 blockade therapy and continued until death, last contact, or 15 March 2022.

Overall survival was calculated from the date of PD-1/PD-L1 blockade therapy initiation to the time of death (from any cause) or the last follow-up visit.

### 2.6. Statistical Analysis

Differences in the continuous variables were assessed using the Mann–Whitney U test, and differences in the categorical variables were compared using the Chi-square test or Fisher’s exact test. Overall survival curves were plotted using Kaplan–Meier analysis, and differences were compared using the log-rank test.

The cytokine levels were compared between cachectic and non-cachectic patients and described using volcano plots. The statistical significance and fold-change of the cytokine expression ratio between cachectic patients and non-cachectic patients were plotted on the y and x axes, respectively. All statistical analysis was conducted with GraphPad Prism 9.3.1 (Dotmatics, San Diego, CA, USA). Two-sided *p*-values lower than 0.05 denoted statistically significant differences.

## 3. Results

### 3.1. Patient Characteristics

From July 2019 to December 2020, a total of 92 patients with advanced lung cancer were treated by PD-1/L1 blockade. Of them, 57 patients who initially received PD-1/L1 blockade therapy participated in the study. Two patients without a record of body weight prior to PD-1/PD-L1 blockade therapy were excluded from the study. Overall, 55 patients were analyzed for cachexia status in this study. Among them, cytokine assay was conducted in 41 patients who had peripheral blood samples both at baseline and during treatment with PD-1/PD-L1 blockade, including 21 patients with cachexia (51.2%) and 20 non-cachectic patients (48.8%) (Appendix A).

In total, 41 men and 14 women were included in the study cohort: 55 patients with a median age of 67 years. Adenocarcinoma was diagnosed in 37 (67.3%). There were 19 patients with PD-L1 ≥ 50% and 30 patients with PD-L1 < 50%. Among these 55 patients, 2 received nivolumab, 35 received pembrolizumab, and 18 received atezolizumab (Table 1).

During immunotherapy, at least one irAE occurred in 45 (81.8%) patients, of whom 19 had severe (≥Grade 3) irAEs. After PD-1/PD-L1 blockade treatment in 55 patients, one patient and 16 patients achieved CR and PR, respectively. Twenty-three patients and nine patients experienced SD and PD, respectively. In addition, the best response to PD-1/PD-L1 blockade therapy was not evaluable in six patients. The median follow-up time was 16.5 months and a total of 23 deaths (41.8%) were recorded in the two years after immunotherapy.

### 3.2. Identification of Cachexia-Related Cytokines

Forty-one patients had peripheral blood samples taken at baseline and during treatment with PD-1/PD-L1 blockade. These were analyzed for circulating inflammatory cytokines. The clinical characteristics of 21 patients with cachexia and 20 non-cachectic patients are presented in Table 2. Compared to non-cachectic patients, patients with cachexia had lower baseline weight and plasma albumin before PD-1/PD-L1 blockade treatment (*p* = 0.039, *p* = 0.013, respectively). However, the NLR, indicative of systemic inflammation, was likely to be higher in patients with cachexia in comparison with non-cachectic patients (*p* = 0.052). No difference in tumor staging, PD-L1 expression, or objective response to PD-1/PD-L1 blockade was observed between patients with cachexia and those without cachexia.

The overall survival was compared between the cachectic patients and non-cachectic patients in a total of 55 patients and 41 patients with cytokine assay, respectively. The overall survival was significantly better in patients without cachexia than in those with cachexia, on comparing both cohorts (two-year survival rate: 70.5% vs. 40.0%, *p* = 0.007; 71.2% vs. 36.7%, *p* = 0.022, respectively) (Appendix A). The patients with a high expression of PD-L1 (≥50%) were likely to have better overall survival than those with low PD-L1 (<50%), although there was no significant difference in their therapeutic responses to PD-1/PD-L1 blockades (Appendix A).

Among 41 circulating inflammatory cytokines, the levels of six cytokines were found to be significantly different between cachectic and non-cachectic patients (Figure 1a,b). Of these six cytokines, IL-6, IL-8, IL-10, and IL-15 at baseline, and IP-10 at both baseline and during treatment were significantly higher in patients with cachexia than those without. In contrast, the baseline eotaxin-1 in patients with cachexia was significantly lower than that in non-cachectic patients (Figure 2).

In addition, the association of PD-L1 expression with cachexia-related cytokines was analyzed among these 41 patients. No significant difference of cachexia-related cytokines was observed between PD-L1 < 50% versus PD-L1 ≥ 50% at baseline and during treatment with PD-1/PD-L1 blockade (Appendix A).

### 3.3. Higher Eotaxin-1 Was Associated with Better Therapeutic Response and Overall Survival

The levels of plasma eotaxin-1 at baseline and during treatment were analyzed based on the therapeutic response to PD-1/PD-L1 blockade therapy and overall survival. There was no significant difference in baseline eotaxin-1 between patients who achieved objective response (CR or PR) and those with SD or PD after PD-1/PD-L1 blockade treatment, although the overall survival tended to be better in patients with higher eotaxin-1 (two-year survival rate: 68.6% vs. 30.6%, *p* = 0.28) (Figure 3a).

As Figure 3b shows, the level of plasma eotaxin-1 after one cycle of PD-1/PD-L1 blockade treatment in patients who obtained CR or PR was significantly higher than those with SD or PD (*p* = 0.039). Consistent with previous better survival rates and higher eotaxin-1 in non-cachectic patients, the overall survival of patients with higher eotaxin-1 was better in comparison with those with lower eotaxin-1 (two-year survival rate: 78.8% vs. 30.6%, *p* = 0.010).

Notably, eotaxin-1 significantly increased after one cycle of PD-1/PD-L1 blockade treatment in all patients with CR or PR compared to patients with SD or PD (*p* = 0.003). A greater increase in eotaxin-1 after one cycle of PD-1/PD-L1 blockade treatment was associated with better overall survival (74.1% vs. 29.3%, *p* = 0.009) (Figure 3c).

### 3.4. Cachexia-Related IL-6 Was Associated with Immune-Related Adverse Events

Higher IL-6 levels, both at baseline and during treatment with PD-1/PD-L1 blockade, were observed in patients experiencing irAEs compared to those in whom irAEs were absent (*p* = 0.0098, *p* = 0.042, respectively). Furthermore, patients experiencing severe irAEs were likely to have higher IL-6 at baseline before PD-1/PD-L1 blockade treatment (*p* = 0.033) (Figure 4a).

There was no significant association between therapeutic response and IL-6 at baseline or during treatment, although patients with higher IL-6 at baseline tended to have worse overall survival (two-year survival rate: 44.9% vs. 61.9%, *p* = 0.14) (Figure 4b,c).

### 3.5. Higher IL-10 at Baseline and during Treatment Predicted Worse Overall Survival after PD-1/PD-L1 Blockade Therapy

There was no significant difference in IL-10 between patients who achieved objective response (CR or PR) and patients with SD or PD. However, higher IL-10 both at baseline and during treatment was associated with poorer overall survival (two-year survival rate: 32.5% vs. 72.6%, *p* = 0.012; 44.3% vs. 62.3%, *p* = 0.109) (Figure 5a,b). This was in line with our previous findings that patients with cachexia were likely to have higher IL-10 and poorer overall survival.

## 4. Discussion

To our best knowledge, this is the first prospective study that longitudinally examined cachexia-related circulating cytokines, both at baseline and during treatment, and assessed their role in predicting the clinical outcomes of lung cancer patients treated with PD-1/PD-L1 blockade. The principal finding was that high IL-6 was associated with cachexia and a higher risk of immune-related adverse events, while high IL-10 was associated with cachexia and poor overall survival. More importantly, our study showed for the first time that an early increase in eotaxin-1 after PD-1/PD-L1 blockade treatment is a favorable factor related to the therapeutic response to PD-1/PD-L1 blockade and overall survival.

The complex mechanisms involved in the development and progression of cancer cachexia have yet to be elucidated and the role of specific cytokine biomarkers has not been clearly identified [17]. Inflammatory cytokines such as IL-1, IL-6, TNF-α, and IFN-γ were commonly identified and associated with the development of cachexia [11]. In the present study, levels of circulating IL-6 and IL-8 were found to be upregulated in cachectic patients. High IL-6 was often noted in previous cancer cachexia studies in comparison with IL-8 [18]. Notably, our study revealed that high IL-6 was associated with anti-PD-1/PD-L1 treatment-related adverse events in patients with lung cancer. This provided direct evidence for the use of anti-IL-6 therapy to treat severe and refractory irAEs induced by PD-1/PD-L1 blockade [19,20]. However, neither circulating IL-8 nor a change in IL-8 levels, which was previously reported to reflect the response to anti-PD-1 treatment, was associated with therapeutic response or irAEs in our study [21].

In our study, we observed a connection between an increase in eotaxin-1 after one cycle of PD-1/PD-L1 blockade treatment, and high therapeutic benefit and improved overall survival. After binding to CCR3 receptors expressed on the cell surface of eosinophils, eotaxin-1 (CCL11) activates a series of intracellular signaling cascades, leading to eosinophil recruitment to inflammatory sites [22,23]. A recent study reported a positive correlation between eosinophil accumulation and CD8+ T cell infiltration in tumor tissues from melanoma patients treated with immune checkpoint blockade [24]. As changes in circulating eotaxin-1 can be detected soon after one cycle of PD-1/PD-L1 blockade treatment, this may be useful for monitoring clinical outcomes in lung cancer patients receiving PD-1/PD-L1 blockade therapy [25].

IL-10 has multiple pleiotropic effects on immunoregulation and inflammation in cancers [26,27]. The IL-10 genotype was reported to correlate with the development of cachexia among patients with gastroesophageal malignancy [28]. In the present study of lung cancer, the upregulation of circulating IL-10 was observed in patients with cachexia and associated with poor overall survival, despite there being no difference in therapeutic responses. Consistently, a high expression of plasma IL–10 was associated with poor survival in multiple cancers [29]. Circulating IL-10 may thus be a valuable biomarker for prognostic prediction and treatment targeting IL-10 [30,31]. More studies are required to clarify the precise roles of IL-10 in immune checkpoint therapy.

One of the limitations of our study is the small number of patients treated with PD-1/PD-L1 blockade. These results should be validated by large-scale population-based studies. In addition, multivariate modeling was not feasible in this setting because of the limited number of patients. Instead, we showed that cachexia-related circulating cytokines may be associated with the prognosis of immunotherapy. Future work is necessary to characterize the mechanism involving eotaxin-1, IL-6, IL-8, IL-10, IL-15, and IP-10.

## 5. Conclusions

We found that an early increase in eotaxin-1 after immunotherapy is a favorable factor related to the therapeutic response to PD-1/PD-L1 blockade and overall survival. High IL-6 was associated with the risk of immune-related adverse events, while high IL-10 was associated with poor overall survival. A blood-based, cachexia-related cytokine network assay may yield potential biomarkers for predicting the clinical response to PD-1/PD-L1 blockade therapy and provide clues for improving the outcome of cachectic patients.

## Figures and Tables

**Figure 1 cancers-15-01170-f001:**
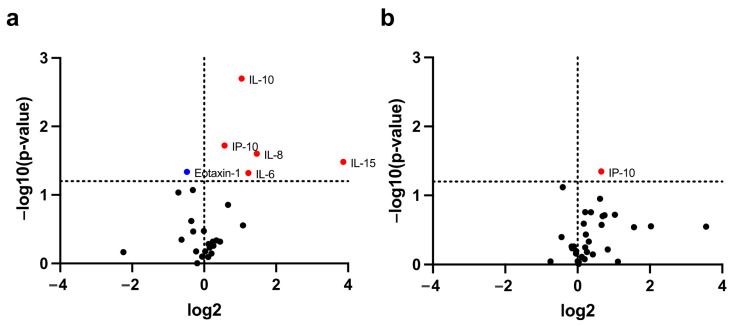
Identification of cachexia-related circulating cytokines. Circulating cytokines with significantly different levels between cachectic and non-cachectic patients at baseline (**a**) and after one cycle of PD-1/PD-L1 blockade treatment (**b**). In the volcano plot, the horizontal dashed line shows where *p* = 0.05, with dots above the line having *p* < 0.05 and dots below the line having *p* > 0.05. The vertical dotted line represents no difference in cytokine levels between cachectic and non-cachectic patients. The blue dot represents a low cytokine level in a cachectic patient, whereas the red dots represent high cytokine levels in patients with cachexia.

**Figure 2 cancers-15-01170-f002:**
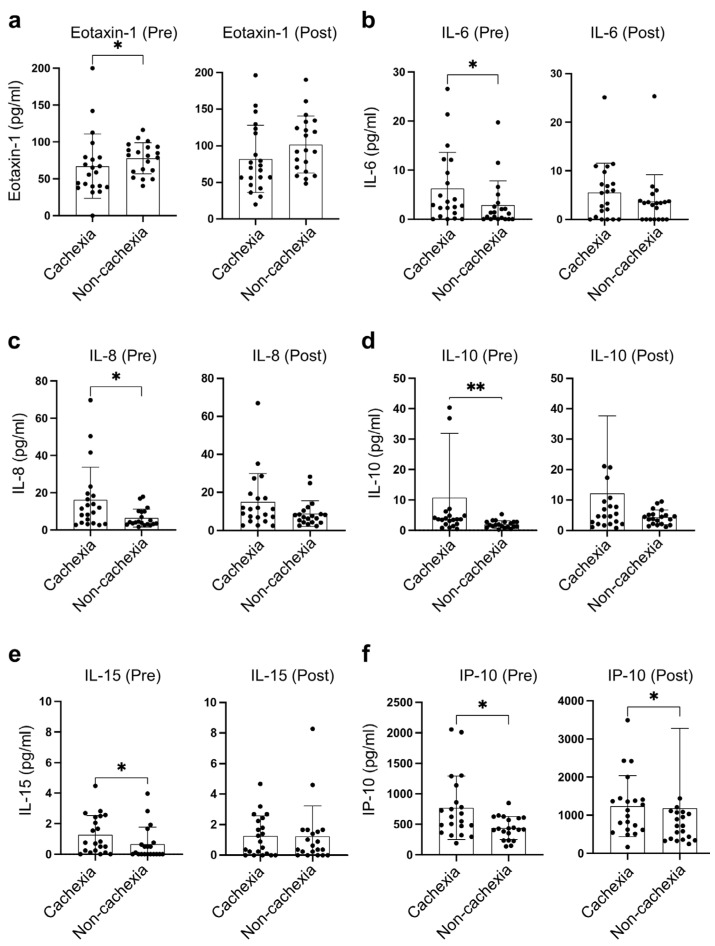
Distinct distribution of cachexia-related cytokines. (**a**) Eotaxin-1, (**b**) IL-6, (**c**) IL-8, (**d**) IL-10, (**e**) IL-15, (**f**) IP-10 between cachexia (n = 21) and non-cachexia (n = 20) at baseline and during treatment with PD-1/PD-L1 blockade. * indicates *p* < 0.05, ** indicates *p* < 0.01.

**Figure 3 cancers-15-01170-f003:**
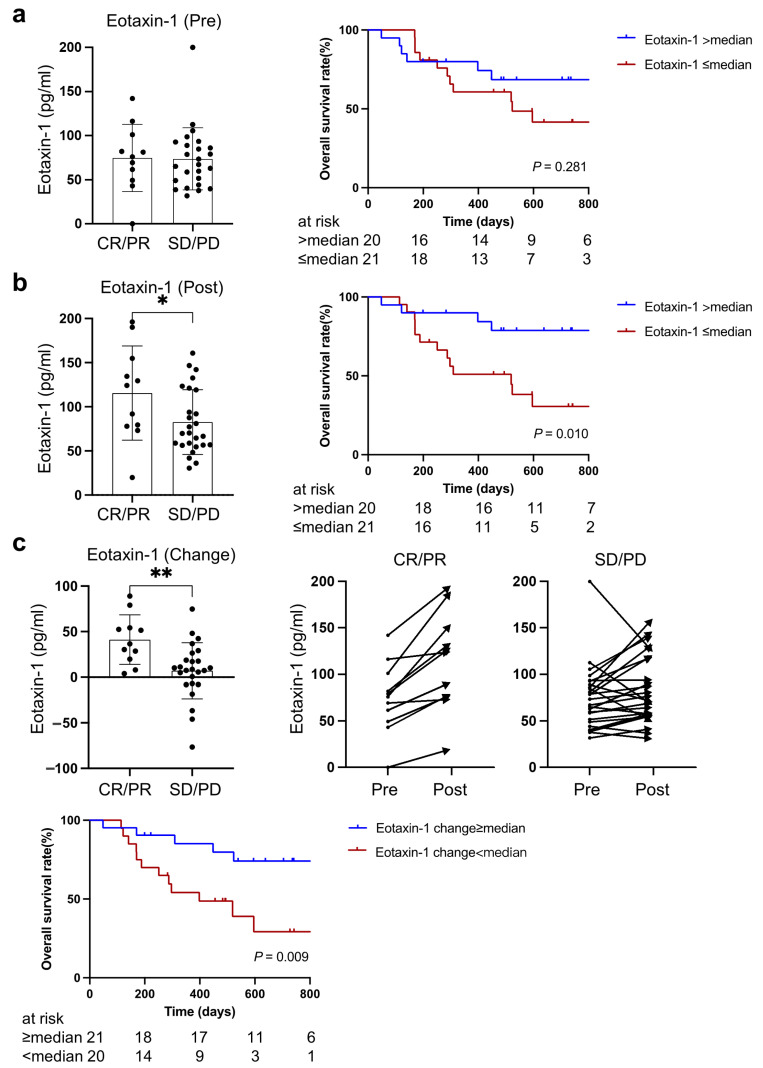
Association of eotaxin-1 with PD-1/PD-L1 blockade treatment outcomes including therapeutic responses and overall survival. (**a**) Baseline level of eotaxin-1, (**b**) eotaxin-1 levels during treatment, (**c**) change in eotaxin-1 levels after three-week PD-1/PD-L1 blockade therapy. CR, complete response; PR, partial response; SD, stable disease; PD, progressive disease. * indicates *p* < 0.05, ** indicates *p* < 0.01.

**Figure 4 cancers-15-01170-f004:**
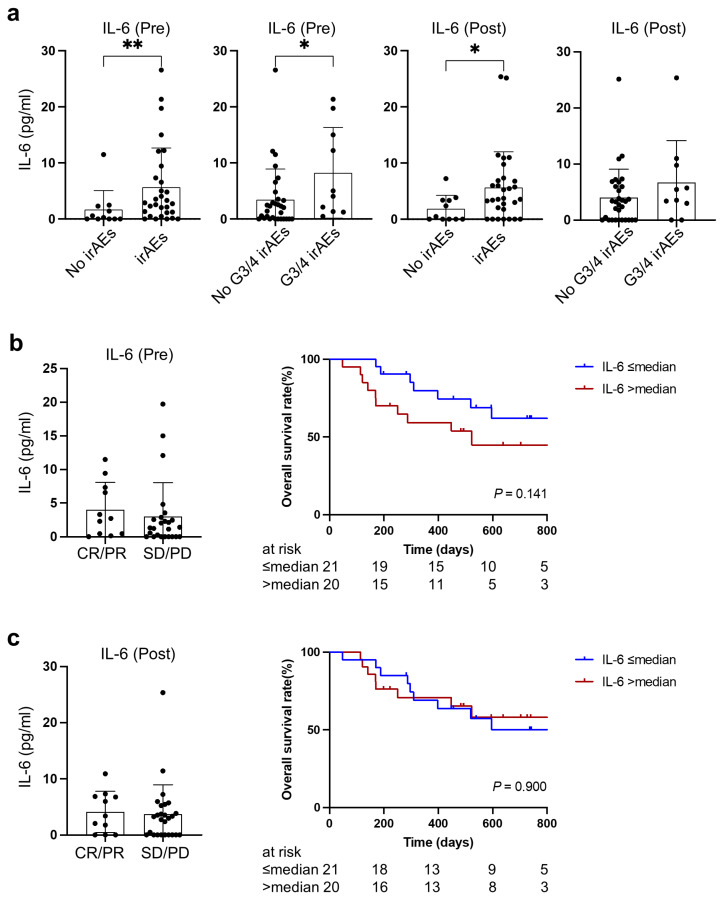
Relationship between IL-6 and therapeutic outcomes at baseline and during treatment. (**a**) Association of IL-6 with irAEs at baseline and during treatment. (**b**) Association of baseline IL-6 with therapeutic response and overall survival. (**c**) Association of IL-6 during treatment with therapeutic response and overall survival. No irAEs (n = 11), irAEs (n = 30), No G3/4 irAEs (n = 31), G3/4 irAEs (n = 10). irAEs, immune related adverse events; CR, complete response; PR, partial response; SD, stable disease; PD, progressive disease. * indicates *p* < 0.05, ** indicates *p* < 0.01.

**Figure 5 cancers-15-01170-f005:**
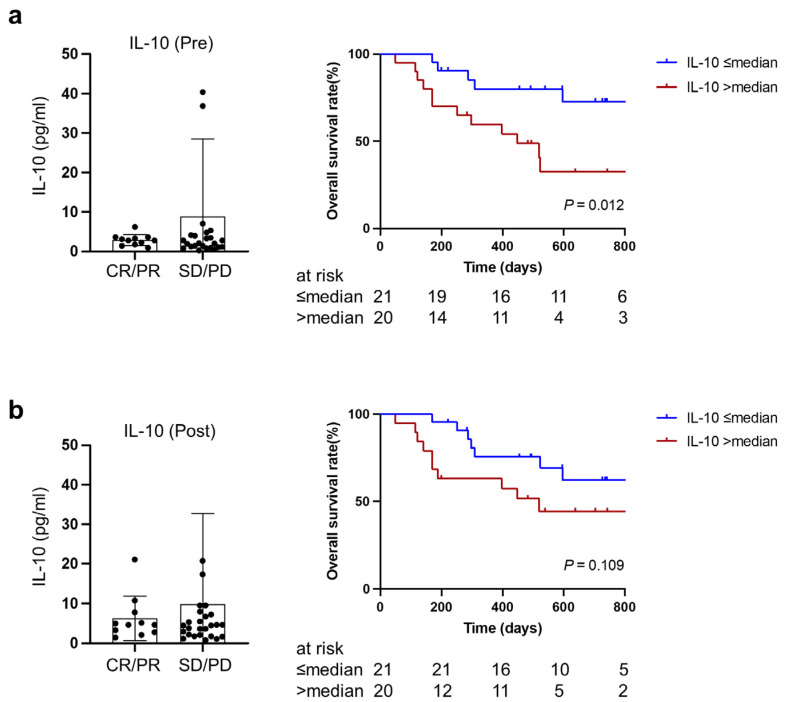
Association of IL-10 at baseline (**a**) or during treatment (**b**) with therapeutic responses and overall survival. CR, complete response; PR, partial response; SD, stable disease; PD, progressive disease.

**Table 1 cancers-15-01170-t001:** Patient characteristics.

Feature	N	Percentage (%)
Sex		
Female	14	25.5
Male	41	74.5
Age at diagnosis (years)		
<60	13	23.6
60~75	26	47.3
≥75	16	29.1
Histology		
Adenocarcinoma	37	67.3
Squamous carcinoma	8	14.5
Others	10	18.2
Clinical stage		
III	3	5.5
IVA	26	47.3
IVB	26	47.3
ECOG PS		
0–1	48	87.3
2	7	12.7
Smoking status		
Never	5	9.1
Former/Current	50	90.9
PD-L1 expression		
<50%	30	54.5
≥50%	19	34.5
NA	6	10.9
Line of immunotherapy		
First	39	70.9
Second	12	21.8
≥Third	4	7.3
Actionable mutation		
(−)/undetected	49	89.1
EGFR(+)	5	9.1
ROS1(+)	1	1.8
Number of metastatic sites		
<3	43	78.2
≥3	12	21.8
irAEs		
No	10	18.2
Grade 1–2	26	47.3
Grade 3–4	19	34.5
Type of immunotherapy		
Nivolumab	2	3.6
Pembrolizumab	35	63.6
Atezolizumab	18	32.7

ECOG PS, Eastern Cooperative Oncology Group Performance Status; PD-L1, Programmed cell death (ligand) 1; EGFR, epidermal growth factor receptor; ROS1, ROS Proto-Oncogene 1; irAEs, immune-related adverse events. NA, not available.

**Table 2 cancers-15-01170-t002:** Clinical characteristics between cachexia and non-cachexia.

Variables	Total (n = 41)	Cachexia (n = 21)	Non-Cachexia (n = 20)	*p*-Value
Sex, male/female	30/11	16/5	14/6	0.734
Age at diagnosis (Y), median (Q1, Q3)	68 (59, 75)	68 (61, 78)	68 (58.5, 74.3)	0.497
Weight at baseline, Kg	61.6 (37.6, 95.6)	55.9 (37.6, 79.0)	62.3 (39.2, 95.6)	0.039
Smoking (smoker/no smoker)	38/3	21/0	17/3	0.107
ECOG PS (0–1/2)	37/4	17/4	20/0	0.107
CRP, mg/dL	2.53 (0.05, 22.68)	3.42 (0.05, 22.68)	1.60 (0.05, 8.25)	0.167
Hb, g/dL	12.7 (7.9, 18.8)	12.3 (7.9, 18.8)	13.2 (9.9, 17.1)	0.105
Alb, g/dL	3.5 (1.9, 4.3)	3.3 (1.9, 4.3)	3.7 (2.8, 4.3)	0.013
NLR	6.02 (1.28, 24.24)	8.09 (1.55, 24.24)	3.85 (1.28, 7.07)	0.052
Histology (ADC/no ADC)	28/13	13/8	15/5	0.506
Stage (III/IV)	2/39	2/19	0/20	0.488
PD-L1expression (<50%/≥50%) *	22/15	10/10	12/5	0.315
irAEs (G3,4 irAEs/no G3/4 irAEs)	10/31	5/16	5/15	>0.99
Objective response (CR, PR/SD, PD) #	11/26	6/12	5/14	0.728

Q1, First quartile; Q3, third quartile; ECOG PS, Eastern Cooperative Oncology Group Performance Status; CRP, C-reactive protein; Hb, hemoglobin; Alb, albumin; NLR, neutrophil-to-lymphocyte ratio; ADC, adenocarcinoma; PD-L1, Programmed cell death (ligand) 1; irAEs, immune-related adverse events; CR, complete response; PR, partial response; SD, stable disease; PD, progressive disease. * NE = 4, # NE = 4.

## Data Availability

The datasets used in this manuscript are available upon request.

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
