# Peer review of "Early Detection of Therapeutic Benefit from PD-1/PD-L1 Blockade in Advanced Lung Cancer by Monitoring Cachexia-Related Circulating Cytokines"

_cancers, 2023, doi:10.3390/cancers15041170_

Round 1

Reviewer 1 Report

The authors described the potential utilities of measuring circulating cytokines for early detection of PD-1/PDL-1 blockade therapy in advanced-stage lung cancer. Overall, the manuscript was well-written, and the discussion was sound. 

Major:

1. Were there any differences in cytokine levels between patients treated with PD-1 blockade and those with anti-PDL-1 as the different blocking targets may lead to different immunological effects?

2. The different expression level of PDL-1 (documented in Table 1 as <50 or >=50) might affect the treatment outcome (CR/PR vs. SD/PD), and the immune responses reflected by the cytokine levels might be also influenced by the PDL-1 expression status. Please comment.

Minor:

1. In Fig 2, 3, 4, and 5, the authors need to specify the units of Y-axis (eg, pg/ml etc.) as they are plasma levels. 

2. The use of volcano plots needs to be described in the method section.

3. This reviewer wonders if the Fig 6 is necessary. This manuscript is an original paper, but not a review one.

Reviewer 2 Report

I do have only one comment.

In my opinion, there is no need of demonstrating the impact of cancer cachexia on overall survival. It is very much understandable, that patients with cachexia shows lower overall survival rate than those non-cachectic. Despite the abovementioned issue, I find this paper to be very well written, therefore I recommend it for further publication processes.

Reviewer 3 Report

Very well written and informative paper on the possible role of cytokines as predictors for response to immunotherapy, specifically related to cachexia cytokines.  A particular strength of the study is the prospective design. Limitation is a rather small number (41) avialable for analysis. Methods are well described and definitions conventional. The results are presented in a very clerar way and discussion is well balanced between significance of findings and study limitations.

My only question is related to selection bias. The authors claim that all patient were recruited but this need to be better described: How many were seen at the clinic, how many were invited to participate, how many were included.

I have no other specific suggestions. 
